# Depth of Cure, Hardness, Roughness and Filler Dimension of Bulk-Fill Flowable, Conventional Flowable and High-Strength Universal Injectable Composites: An In Vitro Study

**DOI:** 10.3390/nano12121951

**Published:** 2022-06-07

**Authors:** Francesco Saverio Ludovichetti, Patrizia Lucchi, Giulia Zambon, Luca Pezzato, Rachele Bertolini, Nicoletta Zerman, Edoardo Stellini, Sergio Mazzoleni

**Affiliations:** 1Department of Neurosciences–Dentistry Section, Università degli Studi di Padova, 35121 Padova, Italy; patrizia.lucchi@unipd.it (P.L.); giulia.zambon@studenti.unipd.it (G.Z.); edoardo.stellini@unipd.it (E.S.); sergio.mazzoleni@unipd.it (S.M.); 2Department of Industrial Engineering, Università degli Studi di Padova, 35121 Padova, Italy; luca.pezzato@unipd.it (L.P.); rachele.bertolini@unipd.it (R.B.); 3Department of Pediatric Dentisrty, Università degli Studi di Verona, 37129 Verona, Italy; nicoletta.zerman@univr.it

**Keywords:** dental materials, light curing, pediatric dentistry

## Abstract

(1) Objective: To evaluate and compare the depth of cure (DOC) of two bulk-fill flowable composites (Filtek Bulk Fill Flowable Restorative and Tetric EvoFlow Bulk Fill), two conventional flowable composites (Filtek Supreme XTE Flowable Restorative and G-ænial Flo X) and one high-strength universal injectable composite (G-ænial Universal Injectable). (2) Methods: specimens were placed in a stainless-steel mold with an orifice of 4 mm in diameter and 10 mm in depth and light-cured for 20 s using a light emitting diode (LED) light-curing unit (LCU) with an irradiance of 1000 mW/cm^2^; depth of cure was assessed using the ISO 4049 scrape technique, and the absolute length of the specimen of cured composite was measured in millimeters with a digital caliper. The same procedure was repeated with 14 samples for each material under investigation, for a total number of 70 test bodies. Material roughness and hardness results were also investigated using, respectively, a 3D laser confocal microscope (LEXT OLS 4100; Olympus) at ×5 magnification and a Vickers diamond indenter (Vickers microhardness tester, Shimadzu^®^, Kyoto, Japan) under 10-N load and a 30 s dwell time. SEM images at 3000 and 9000 magnification were collected in order to study the materials’ filler content. Statistical analysis were performed by a commercial statistical software package (SPSS) and data were analyzed using multiple comparison Dunnett’s test. (3) Results: The average DOC of both bulk-fill composites was more than 4 mm, as a range of 3.91 and 4.53 mm with an average value of 4.24 and 4.12 mm, while that of the conventional flowable composites was much lower, as a range of 2.47 and 2.90 mm with an average value of 2.58 and 2.84 mm; DOC of the high-strength injectable composite was greater than the one of traditional composites, but not to the level of bulk-fill materials, as a range of 2.82 and 3.01 mm with an average value of 3.02 mm. Statistical analysis revealed significant differences (*p*-values < 0.05) in the depth of cure between bulk fill flowable composites and other composites, while there was no difference (*p*-values > 0.05) between the materials of the same type. (4) Conclusions: Bulk-fill flowable composites showed significantly higher depth of cure values than both traditional flowable composites and high-strength injectable composites.

## 1. Introduction

The last few decades have seen a rapid and steady increase in the use of resin-based composite (RBC), due to their good biocompatibility in the mouth, esthetics, non-toxicity characteristics compared to older filler materials and their good physical and mechanical properties [1,2]. Clinical success of a RBC’s restoration depends on several factors, including depth of cure, degree of conversion, shrinkage stress and others. Depth of cure (DOC) can be defined as the thickness of resin that can be converted from monomer to polymer [3]. DOC is related to many factors such as the filler size, type and content, the thickness and shade of the material and the effectiveness of light transmission (irradiance, exposure time, distance of the light source) [4]. Inadequate polymerization is significantly associated with a decrease in the physical and biocompatibility properties of RCBs [4]. Anyway, the main cause of failure of RBC restorations is the shrinkage generated during the polymerization reaction of the material [5]. In clinical practice, this process can lead in the medium to long term to secondary caries and restoration failure [6]. Many practical methods have been proposed to reduce shrinkage stress, such as an accurate control of curing light irradiance [7], the use flowable resin liners [8] and incremental layering techniques [9,10]. Literature is almost concordant in defining the maximal increment thickness of conventional RBC as 2 mm, to ensure adequate conversion of the unpolymerized resin [11,12]. This requirement of incremental placement technique in cavities deeper than 2 mm complicates the placement of conventional RBC, with the risk of incorporating air bubbles, creating voids, contamination and bond failures between adjacent layers [13,14].

Therefore, a class of composites has been developed in the last decade, so-called “bulk-fill” materials, which are designed to be placed in 4 or 5 mm thick increments to be cured in one step. These materials can overcome the time consuming incremental layering technique with low polymerization stress and excellent physical properties in terms of wear, functionality and esthetics [15]. Thanks to these properties, bulk-fill RBCs are also promoted as restorative materials particularly suitable for patients with limited compliance, such as pediatric patients and ones with dental anxiety. In order to achieve these characteristics and to improve polymerization depth, manufacturers have developed several strategies. These include a change in photoinitiators systems, which are alternative and more reactive and in the filler, reduced in percentage and larger in size to increase the translucency of the material [16,17]. In addition, some chemical modifications have been made to the monomers, such as increasing the molecular weight, adding new stress-relieving monomers and incorporating methacrylate monomers containing a third reactive site [18]. The use of bulk-fill is still growing and it is not clear that they are completely effective or have the potential to replace traditional composites; they are commonly categorized into high-viscosity composite (sculptable, full-body), usually used to fill up the whole cavity and sculpt the occlusal surface and low-viscosity composite (flowable, base), used as dentine substitutes and often in combination with a high viscosity material [17]. In addition, there are alternative dispensing systems using ultrasonic waves, commercially known as SonicTM fill and SonicTM fill 2 (Kerr Corporation) [12].

Since bulk-fill composites use in clinical practice is emergent, the confirmation of their effectiveness and their potential to replace traditional composites is still uncertain. The aspects that have been most questioned mainly concern the appropriate polymerization in the thicknesses indicated by the manufacturers. This, if not sufficient, may decrease the mechanical and biocompatibility properties of the restoration, as well as its survival over time. In the last few years, in vitro studies about depth of cure (DOC) and other clinical performances have been increasing substantially [19]. Although several studies have confirmed the depths indicated by the manufacturers, with a 4 mm DOC, others have reported conflicting results; hence, to this day, there is a lack of homogeneity in the literature on different bulk-fill composites [15,20,21,22].

The aim of the present study is to evaluate and compare the depth of cure of two bulk-fill flowable composites, two conventional flowable composites and one high-strength universal injectable composite, using the International Organization for Standardization (ISO) 4049 scrape technique. The null hypothesis is that there is no statistical difference between the DOC of the materials under investigation.

## 2. Materials and Methods

This in vitro study was conducted in UOC Clinica Odontoiatrica, Neuroscience Department, Università degli Studi di Padova, Italy. The study was performed using five different RCBs materials: two bulk-fill flowable materials, Tetric EvoFlow Bulk Fill (Ivoclar Vivadent) and Filtek Bulk Fill Flowable Restorative (3M), one high-strength injectable composite, G-ænial Universal Injectable (GC), two conventional flowable composites, Filtek Supreme XTE Flowable Restorative (3M) and G-ænial Flo X (GC). Material descriptions, manufacturers and composition are listed in Table 1.

### 2.1. Depth of Cure

Depth of cure of the investigated materials was determined according to the International Organization for Standardization (ISO) 4049 method [23]. 

A reusable cylindrical stainless steel mold with an orifice of 4 mm in diameter and 10 mm in depth was prepared (Figure 1). The mold was placed on a mylar strip and the hole was entirely filled with one of the studied materials, taking care of fully fill the internal space and avoiding the creation of possible voids. The top side of the mold was covered with a second mylar strip and excess material was pressed out. The composite was then light-cured for 20 s using a VALO (Ultradent) LED light-curing unit (LCU) with an irradiance of 1000 mW/cm^2^ (according to manufacturers’ recommendations), making sure to keep the light tip centered and in contact with the material. After light-curing, the cylindrical specimens (Figure 2) were pushed out of the stainless steel mold by using a stainless steel “pin driver” (Beta) with a diameter equal to the mold hole and the uncured resin composite material was gently scrapped off with a plastic spatula.

The absolute length of the cylindrical specimen of cured composite was then measured in millimeters with a digital caliper of ±0.1 mm accuracy (Qfun) (Figure 3), for a total of three measurements for each single specimen. The absolute lengths were divided by two and these values were recorded as ‘Depth of Cure’ (DOC). The same procedure was repeated with 14 samples for each material under investigation, for a total number of 70 test bodies. The procedure was performed by a single operator in order to eliminate variation in the applied force when scrapping the materials. Statistical analysis was performed by a commercial statistical software package (SPSS). 

The results were analyzed using Dunnett’s test, a multiple comparison test used to compare each of a number of treatments with a single control. This test was used to test the significance of difference between the 5 groups; a *p*-value of less than 0.05 is taken to denote the significant relationship. According to Pena and Mingoti, the Dunnet’s test keep estimates of the probability of type I error similar and close to 0.05. In addition, Dunnett’s and Student’s t tests showed similar test power estimates with a slight advantage to Dunnett’s test.

### 2.2. Roughness

The surface roughness (Ra) was measured by using the PLU-Neox™ optical profiler (Sensofar, Barcelona, Spain) equipped with a 20× confocal objective (Nikon™, Tokyo, Japan) with a 0.45 numerical aperture. An area of 1.6 × 0.6 mm^2^ was acquired for the surface of each group; afterward, eigh surface profiles were extracted to compute the roughness according to ISO 4288, with λs filter equal to 2.5 μm, cut-off λc equal to 0.8 mm. An example of the cropped surface texture and relative surface roughness profile is shown in Figure 4 and Figure 5 [24]. 

### 2.3. Microhardness

For the microhardness test (*n* = 5), 5 indentations were made in each specimen using a Vickers diamond indenter (Vickers microhardness tester, Shimadzu^®^) under 10 N load and a 30 s dwell time. Hardness values (GPa) were calculated according to the equation H = P/2d^2^, where P is the load in newtons and d is the average of the diagonal values [25]. 

For both roughness and microhardness, the results were analyzed using Dunnett’s test, a multiple comparison test used to compare each of a number of treatments with a single control. This test was used to test the significance of difference between the 5 groups; a *p*-value of less than 0, 1 is taken to denote the significant relationship.

### 2.4. Scanning Electron Microscope

Morphological characteristics of samples were studied using the scanning electron microscope SEM (Leica Microsystems s.r.l., Milan, Italy). To be observed correctly, samples must be conductive, so a metallic layer should be placed on top of them. In detail, samples in this study were coated with a gold layer of approximately 20 nm. The microscope used for the experimental activity was a Cambridge Stereoscan 440 (Leica Microsystems s.r.l., Milan, Italy), equipped with a Philips PV9800 EDS (Leica Microsystems s.r.l., Milan, Italy) microanalysis available at the Industrial Engineering Department of the University of Padua. Images were taken using the secondary electron detector [24,25]. 

## 3. Results

### 3.1. Depth of Cure

First of all, descriptive analyses were carried out, using a box-plot diagram (Figure 6), where the distribution of the measurements is graphically represented by means of scatter and position indices. The mean, median, standard deviation, range and coefficient of variation (%) were also calculated for each group. In Table 2, we can notice how Group 1 and Group 2 present a deeper depth of cure compared to others, with an average DOC of 4.24 mm in a range between 3.91 and 4.53 mm for G1 (Filtek Bulk Fill Flowable Restorative) and a mean value of 4.12 mm in a range between 3.91 and 4.32 mm for G2 (Tetric EvoFlow Bulk Fill). Concerning other materials, we notice that they present lower average DOCs, which were 3.02 mm in a range between 2.82 and 3.15 mm for G3 (G-ænial Universal Injectable), 2.58 mm in a range between 2.47 and 2.75 mm for G4 (Filtek Supreme XTE Flowable Restorative) and 2.84 mm in a range between 2.76 and 2.90 mm for G5 (G-ænial Flo X); G4 was the one with the lowest values, and G3 was the group with more outlier points. Additionally, the median of each group is very close to the mean of the respective group: this indicates symmetry.

Statistical analysis was performed through Dunnett’s multiple comparison test. Corresponding *p*-values for each test are described in Table 3. These analyses revealed a significant difference (*p*-values < 0.05) in the depth of cure between the two bulk fill flowable composites (Group 1-2) and other materials (Group 3-4-5). Furthermore, there is no statistically significant difference between G1 and G2 (*p*-value 0.6461) and between G3-G5 (*p*-value 0.2348) and G4-G5 (*p*-value 0.1687), showing homogeneity between different materials of the same type. 

### 3.2. Roughness

In Figure 7, we observe the Boxplot of the 5 treatments considered in this study; it can be noted that the G2 group differs from the other groups in relation to the roughness found. The other groups have relatively similar behavior. Among all groups, the G2 group has greater variability than the others. As for the G5, we noticed that this was the one that featured the lowest values while the G4 and G1 featured outliers points.

In Table 4 we observe the main descriptive measures of each group with regard to roughness. We see that the G2 mean is well above the others, the G3 and G5 groups have similar averages. The largest amplitude (range), i.e., the difference between the minimum and maximum observed value, is in group 2, while the smallest is in group 3.

Statistical analysis was performed through Dunnett’s multiple comparison test. Corresponding *p*-values for each test are described in Table 5. G2 is statistically different when compared with the G3 and G5 groups.

G4 obtained, on average, a higher roughness value than groups 3 and 5.

Furthermore, group 5 was significantly different in relation to group 1 and in this case group 1 had a greater roughness than group 5. Finally, looking at the *p*-values among the other groups, we noticed that they were greater than 0. 10, so we did not reject the null hypothesis, i.e., there was no significant difference in the roughness found.

### 3.3. Microhardness

In Figure 8, we have the xox-plot of the five treatments considered in this study, and in it we can see that the G2 group stands out from the other groups in relation to the hardness found, being higher. Among all groups, groups G1 and G5 have greater variability than the others. As for G4, we noticed that this was the one that featured the lowest values while G2 and G4 featured outliers points. The other groups showed relatively similar behavior to each other.

In Table 6, we observe the main descriptive measures of each group. We see that the average hardness of G2 is well above the others, furthermore, we realize that the G4 and G5 groups have very close averages as well as G1 and G3. The median of each group is very close to the mean of the respective group, this indicates a certain symmetry between the data. The largest amplitude (range), i.e., the difference between the minimum and maximum observed value, is in group 1, while the smallest in group 3.

Statistical analysis was performed through Dunnett’s multiple comparison test. Corresponding *p*-values for each test are described in Table 7. the G2 treatment presents a significant difference with groups G4 and G5, i.e., looking at this result together with graph 1 we find that G2 obtained, on average, a higher hardness value than groups 4 and 5, however, there was no significant difference between group 2 compared to groups 1 and 3.

When we look at G4, it is statistically different when compared to groups 3, so G4 got, on average, a lower value than groups 3. Finally, when we look at the *p*-values between the other groups, they were greater than 0.10, so we did not reject the null hypothesis, i.e., there was no significant difference in the hardness found.

### 3.4. Scanning Electron Microscope (SEM)

SEM images can be seen in Figure 9. In groups 4 and 5 the size of the filler is micrometric. In groups 1, 2 and 3 the size of the filler is close to one micron in size. Group 3 has filler dimensions similar to those of groups 1 and 2, but the filler seems better distributed since no empty areas are seen. In general, we can see that in “bulk” materials the filler percentage is lower, but larger in size than in “non-bulk” materials. These findings confirm what is expected, since a smaller amount of filler favors the passage of photopolymerizing light and therefore allows a deeper polymerization of the material.

### 3.5. Figures, Tables and Schemes

Depth of cure is defined as the thickness of resin that can be converted from monomer to polymer under a specific light-curing condition [3]. This is the key parameter for positioning material increments without compromising the physical and biological properties of composite materials. Inadequately cured composites show reduced physical and mechanical properties and can be cytotoxic to the dental pulp due to the increased content of free monomers [25]. In the present study, depth of cure of the different materials was found using the scraping method described in ISO 4049. Several other methods are available for testing depth of cure, including Vickers microhardness test and infrared spectroscopy. ISO 4049 has been accused to overestimating the depth of cure, as shown in the studies by Flury et al. and Moore et al. [26,27] However, this method was chosen because it is the most commonly used due to its simplicity and cost-effectiveness, subjecting different test materials to the same curing conditions within a single study [19]. Bulk fill flowable materials were chosen because they are particularly useful in the clinic, due to their qualities of reduced working time and easy handling, especially in cases of pediatric and uncooperative patients. They were chosen to be compared with traditional flowable composites and high-strength composites for a comparative purpose.

The values were found to be highly variable between the different composite types, with increased thicknesses for both bulk-fill flowable materials (G1-G2) than for the two control materials (G4-G5) (Table 3). G1 and G2, i.e., bulk-fill composites, showed significant statistical differences (*p*-values < 0.05) (Table 5) with all other composites tested. Under a 20-s irradiation using an LED unit with an intensity of 1000 mW/cm^2^, the bulk-fill flowable composites tested demonstrated maximum depth-of-cure values consistent with the manufacturers’ specifications, both stated as 4 mm [28,29]. On the other hand, traditional flowable composites recorded average cure depths of 2.58 mm (G4) and 2.84 mm (G5) (Table 3), well below the values of bulk-fill materials, aligning with the widely known recommendations not to exceed a thickness of 2 mm per layer when placing these materials. At the same time, however, no statistically significant difference can be found between G1 and G2 as well as between G4 and G5, showing homogeneity between the various groups (Table 5).

A new product recently introduced to the market, a high-strength nano-filled injectable composite material, GC’s G-ænial Universal Injectable (G3), was also included in the study. This is a low-viscosity composite in which modifications have been made to provide a product with high wear resistance and depth of cure to make it suitable for posterior restorations. This material, with an average DOC of 3,02 mm (Table 3), reported statistically significant differences with both G1–2 and G4, while there were no differences with G5 (Table 5). This suggests that the depth of cure of G3 is not comparable to that of the bulk-fill composites included in the study, but is at the same time greater than the one of some traditional flowable composites.

These results are in agreement with the study by Flury et al. (2012), who found that the ISO 4049 method showed higher depth of cure values for bulk-fill composites than for conventional materials [26]. Other more recent studies, such as those by Garcia et al. (2014) and Anand Yokesh et al. (2017), who investigated the behavior of some bulk-fill composites in their fluid composition, led to the same results [22,30]. Bulk-fill RBCs have some differences in their chemical composition compared to conventional RBCs. In Tetric EvoCeram Bulk Fill, the manufacturer states that, in addition to the regular camphorquinone/amine initiator system (CQ), an initiator booster (Ivocerin) has been introduced, in order to increase depth of cure of the material [29]. It is based on benzoyl germanium and classified as type I, which means that it does not require a co-initiator to produce free radicals, resulting in a greater reactivity during light curing. Nevertheless, benzoyl germanium photoinitiator is sensitive at lower wavelengths (between 380 and 450 nm), compared with CQ (450–490 nm), so this alternative photoinitiator may not be excited by monowave LCUs, as they deliver photons only within blue light spectrum (400–500 nm); for these reasons, the usage of polywave LCUs are recommended [18]. On the other hand, Filtek Bulk Fill Flowable Restorative does not specify any changes regarding the initiator, but states that the matrix consists of a combination of four monomers with high molecular weight (BisGMA, BisEMA, Procrylat and UDMA) [28]. However, the most important modification for increasing depth of cure is the improved translucency of bulk-fill RBCs. Translucency of the material is influenced by the amount of filler [31], the difference in refractive indices between the filler particles and the resin matrix [32], which determines how light is scattered within the material [33] and the filler size. Increasing the size of the filler decreases, at a similar amount of filler, the total filler surface and, consequently, the filler-matrix interface, increasing the amount of absorbed light that can activate the photoinitiator. In addition, in nano-filled RBCs there are some particles that are smaller than the wavelength of visible light (390 to 750 nm), which are unable to scatter or absorb visible light, increasing translucency and aesthetics of the RBC [34]. Even material viscosity seems to be an important factor on depth of cure. Flowable bulk-fill RBC have a composition percentage of inorganic filler that ranges between 64 and 75 wt%/38 and 61 vol%, presenting a greater amount of organic matrix, that may produce a reduction on flowable resin composites light refractive index, when compared with high viscosity materials. In addition, their matrix consists of low-molecular weight monomers, that have higher flexibility and reactivity, which permits increased formation of binding sites during light curing, consequently, raising its degree of conversion [18].

Tsujimoto et al. in a 2017 paper reported discordant results with those observed in our study [35]. Their study, using ISO 4049 scraping method, showed a depth of cure by 20-s irradiation of less than 4 mm for all materials they tested, for both high- and low-viscosity bulk-fill composites. For Filtek Bulk Fill Flowable Restorative (3M), for example, they observed an average depth of 3.73 mm. As they increased the exposure time, however, the value rose to 3.95 mm at 30 s and 4.24 mm at 40 s. This is supposed to be due to the fact that in Tsujimoto’s study, a quartz-tungsten halogen-type LCU was used as energy source, with an intensity of 600 mW/cm^2^. In the present study, on the other hand, the energy was provided by a LED LCU with an intensity of 1000 mW/cm^2^. Indeed, it has been widely demonstrated that the effectiveness of light is influenced by many factors including wavelength, light intensity, distance from the light source and exposure time [36,37]. The ISO standard does not expressly specify the type of LCU to be used, but it recommends the use of a source specified by the manufacturers for the materials tested. According to the technical manual for Filtek Bulk Fill Flowable the test bodies can be cured indiscriminately by a halogen LCU with an output of at least 550 mW/cm^2^ or by a high intensity LED light (minimum 1000 mW/cm^2^). However, it is reported that, while the polymerization with a halogen LCU must last at least 40 s, the polymerization time can be halved for all shades by using LED LCU [28]. Therefore, in Tsujimoto’s study the clinically valid result will be the one measured after an irradiation time of 40 s, which is in line with the values observed in our study.

Clinical implications of this study are significant. While conventional composites still proved to be dependent on the classical layering technique, with maximum thickness increments of 2 mm, bulk-fill composites proved to be suitable, in vitro, for the incremental so-called “bulk-filling” technique, in which cavities are filled in a single layer up to 4 mm thick. The use of this technique undoubtedly simplifies the restorative procedure and leads to a considerable saving of clinical time in cases of deep and wide posterior cavities, while also reducing the operator-dependent sensitivity of the technique. This is also reflected in a lower risk of contamination during placement, which decreases the risk of secondary caries. Bulk-fill composites, therefore, especially in the low viscosity composition, due to their special characteristics of reduced working time and easy handling, combined with excellent physical properties especially in terms of depth of cure and shrinkage stress, can represent, in vitro, an advantageous alternative to traditional composites. They may find an important clinical application in pediatric dentistry, where the presence of uncooperative pediatric patients greatly complicates clinical procedures, reflected in a greater risk of inadequate restorations. The reduction of operative time is of crucial importance in these cases and bulk-filled restorations can represent an excellent combination of aesthetics and longevity of adhesive restorations and operative simplicity.

A possible limitation of the study is the shape of the cavity and the use of stainless steel mold, which have a different translucency and reflection from enamel and dentine. However, as proved by Erickson and Barkmeier [38], the depth values measured in white molds are greater than in black or stainless steel molds. In future, a study closer to the clinical scenario with extracted teeth should be required, as well as the investigation of the maintenance of physical properties, especially abrasion, in the medium to long term.

## 4. Conclusions

According to the results, flowable bulk-fill composites (Filtek Bulk Fill Flowable Restorative and Tetric EvoFlow Bulk Fill) show significantly higher depth of cure values and roughness values than both traditional flowable composites (Filtek Supreme XTE Flowable Restorative and G-ænial Flo X) and high-strength injectable composite (G-ænial Universal Injectable). The two bulk-fill RBCs studied (Filtek Bulk Fill Flowable Restorative and Tetric EvoFlow Bulk Fill) do not show any statistically significant difference in depth of cure, while the high-strength universal RBC (G-ænial Universal Injectable) show a depth of cure higher than a traditional flowable RBC (Filtek Supreme XTE Flowable Restorative), but lower than bulk-fill composites under study. Flowable bulk-fill composites (Filtek Bulk Fill Flowable Restorative and Tetric EvoFlow Bulk Fill) and high-strength injectable composite (G-ænial Universal Injectable) show similar results in terms of hardness, which is statistically higher when compared with both traditional flowable composites (Filtek Supreme XTE Flowable Restorative and G-ænial Flo X) hardness.

## Figures and Tables

**Figure 1 nanomaterials-12-01951-f001:**
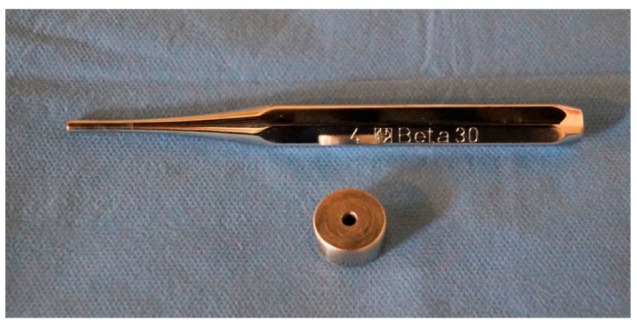
Reusable cylindrical stainless-steel mold with an orifice of 4 mm in diameter and 10 mm in depth. The hole was entirely filled with each of the studied materials, taking care of fully fill the internal space and avoiding the creation of possible voids.

**Figure 2 nanomaterials-12-01951-f002:**
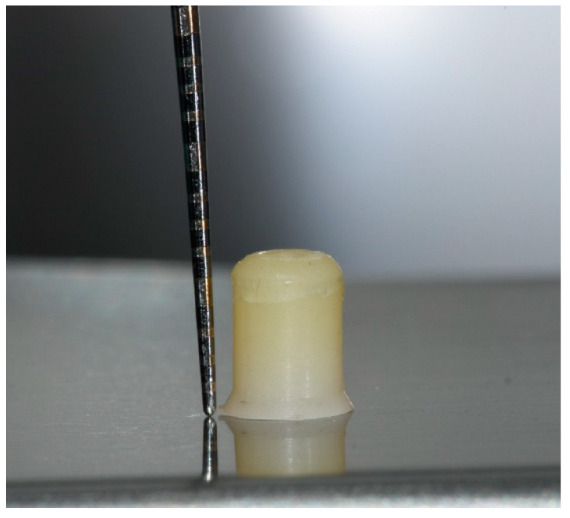
Cylindrical specimens of cured composite pushed out of the stainless steel mold. The uncured resin composite material was gently scrapped off with a plastic spatula.

**Figure 3 nanomaterials-12-01951-f003:**
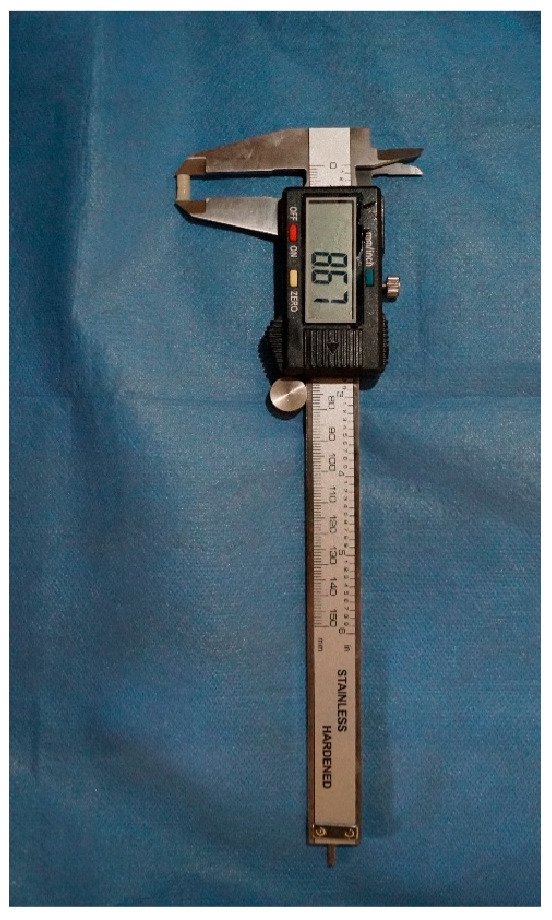
Digital caliber of ±0.1 mm accuracy (Qfun) used to measure the absolute length of the specimen of cured composite.

**Figure 4 nanomaterials-12-01951-f004:**
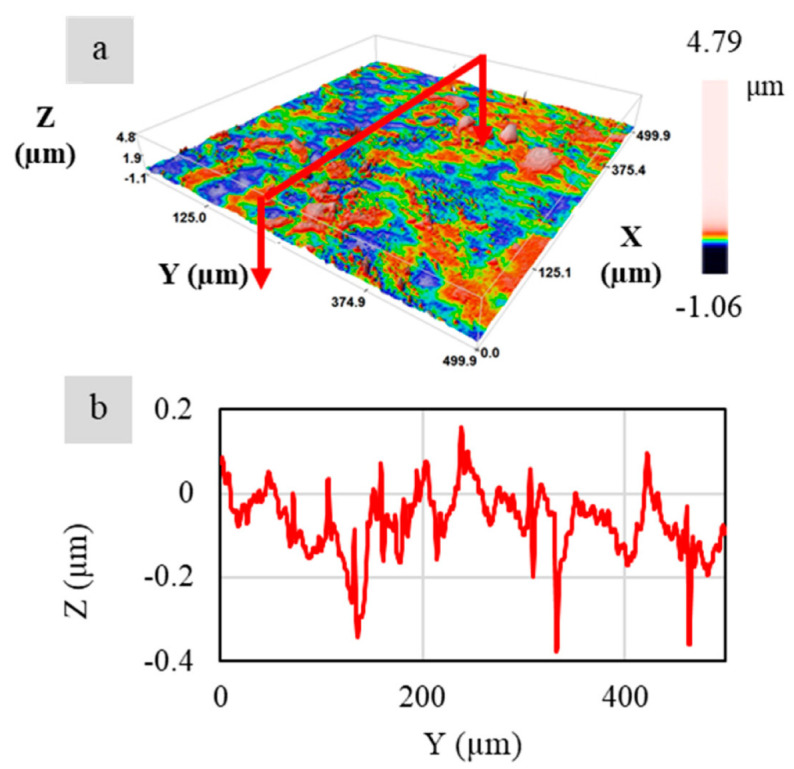
(**a**) Texture. (**b**) Relative surface roughness profile.

**Figure 5 nanomaterials-12-01951-f005:**
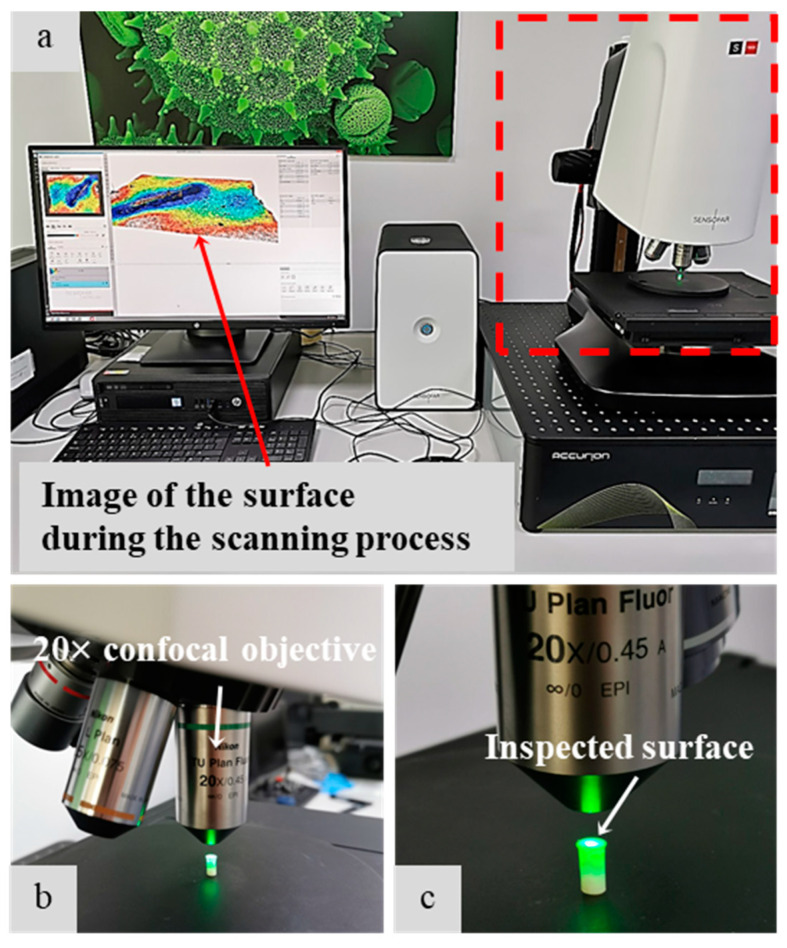
(**a**) Sensofar^TM^ Plu-Neox optical profiler with a zoom on the confocal objective (**b**) and the sample (**c**) during measurements.

**Figure 6 nanomaterials-12-01951-f006:**
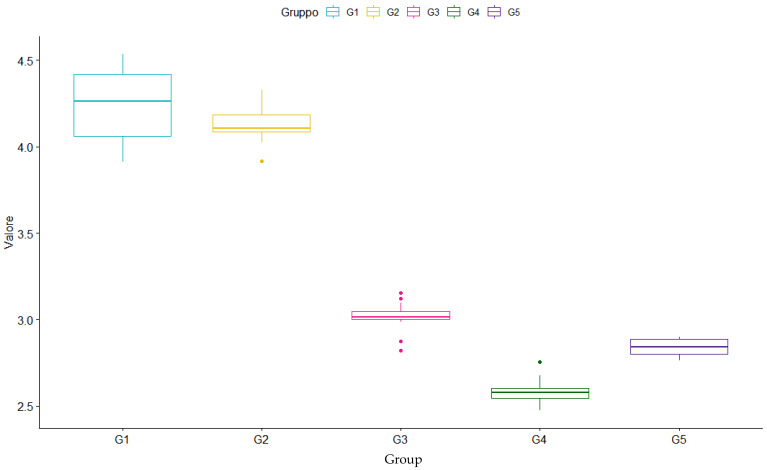
Depth of cure box-plot.

**Figure 7 nanomaterials-12-01951-f007:**
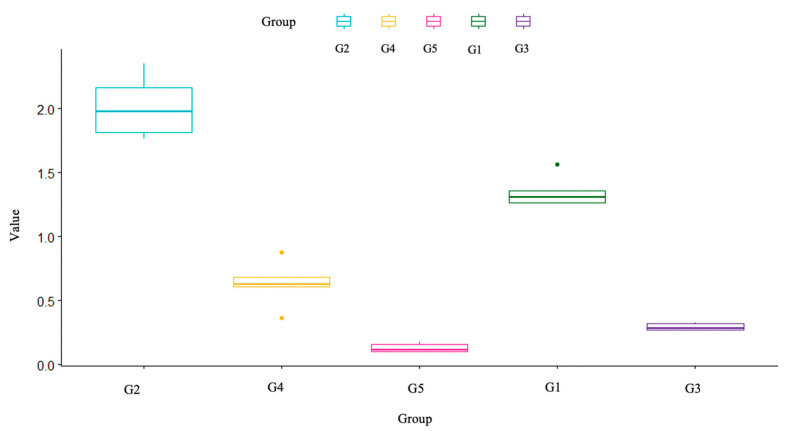
Roughness box-plot.

**Figure 8 nanomaterials-12-01951-f008:**
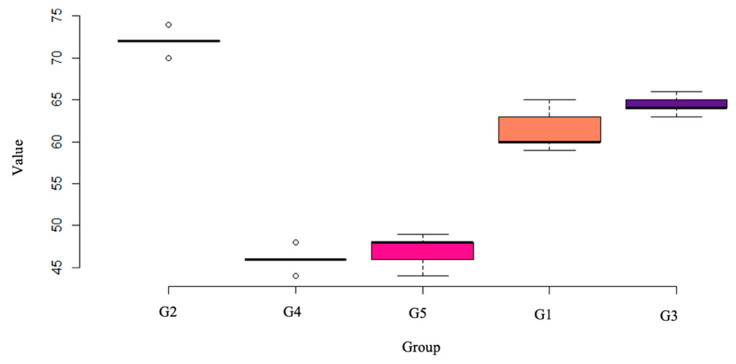
Microhardness box-plot.

**Figure 9 nanomaterials-12-01951-f009:**
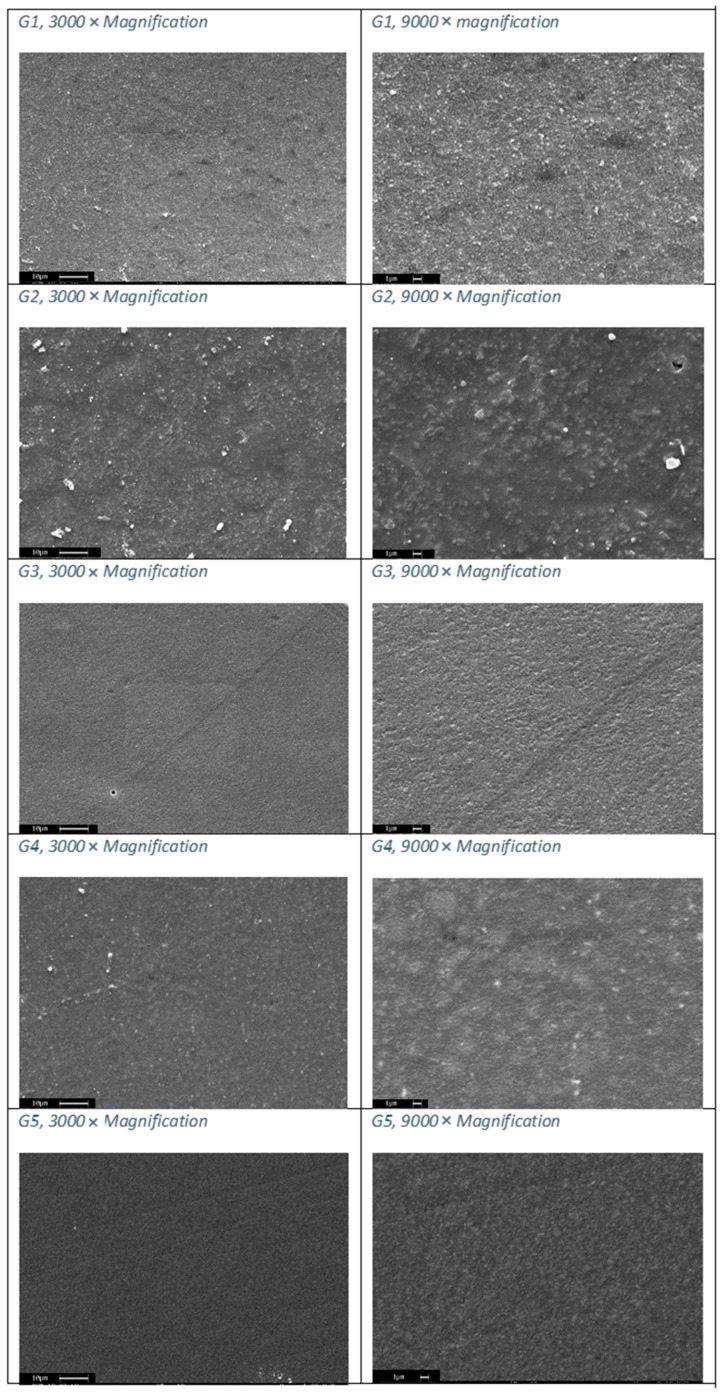
SEM images, 3000 and 9000 magnification.

**Table 1 nanomaterials-12-01951-t001:** Material descriptions, manufacturers and composition.

Group	Material	Composition	Filler % (wt%/vol%)	Type	Manufacturer	Shade	Lot
1	Filtek Bulk Fill Flowable Restorative	Monomers: BisGMA, BisEMA, ;Procrylat, UDMA; filler: zirconia/silica, ;YbF3; photoinitiator: camphoroquinone	64.5/42.5	Flowable bulk fill composite	3M ESPE, Seefeld, Germany	A2	NC36727
2	Tetric EvoFlow Bulk Fill	Monomers: BisGMA, DMA; filler: barium alumino-fluoro-silicate glasses; photoinitiator: Ivocerin	80/60	Nanohybrid flowable bulk fill composite	Ivoclar Vivadent, Schaan, Liechtenstein	IVA	Z017XK
3	G-ænial Universal Injectable	Monomers: dimethacrylate rmonomers; filler: barium glass, silica; photoinitiator	69/50	Nanofilled high-strength low-viscosity composite	GC Europe	A2	201209A
4	Filtek Supreme XTE Flowable Restorative	Monomers: BisGMA, TEGMA, Procrylat; filler: zirconia/silica, YbF3; photoinitiator: camphoroquinone	65/46	Nanofilled flowable composite	3M ESPE, Seefeld, Germany	A2	NC65193
5	G-ænial Flo X	Monomers: UDMA, dimethacrylate monomers; filler: barium glass; photoinitiator	69/50	Microhybrid flowable composite	GC Europe	A2	1811071

BisGMA: Bisphenol A polyethylene glycol diether dimethacrylate, BisEMA: Bisphenol A polyethylene glycol diether dimethacrylate, UDMA: Urethane dimethacrylate, TEGDMA: Triethylene glycol dimethacrylate, BisMEPP: 2,2-bis (4-methacryloxy ethoxy phenyl) propane, YbF3: ytterbium trifluoride.

**Table 2 nanomaterials-12-01951-t002:** Depth of cure descriptive analysis: mean value, standard deviation, median value, minimum and maximum value, range.

Group	Mean	Standard Deviation	Median	Mininum	Maximum	Range
G1	4.24	0.210	4.263	3.913	4.533	0.62
G2	4.12	0.095	4.106	3.918	4.327	0.409
G3	3.02	0.087	3.016	2.822	3.153	0.331
G4	2.58	0.075	2.578	2.473	2.753	0.28
G5	2.84	0.047	2.841	2.765	2.9	0.135

**Table 3 nanomaterials-12-01951-t003:** Depth of cure statistical analysis: multiple comparison test.

Comparison Groups	*p*-Value
G1-G2	0.6461
G1-G3	0.0084
G1-G4	0.0001
G1-G5	0.0006
G2-G3	0.0334
G2-G4	0.0007
G2-G5	0.0002
G3-G4	0.0031
G3-G5	0.2348
G4-G5	0.1687

**Table 4 nanomaterials-12-01951-t004:** Roughness descriptive analysis: mean value, standard deviation, median value, minimum and maximum value, range.

Group	Mean (µm)	sd	Median (µm)	Min (µm)	Max (µm)	Range (µm)
G1	1.351	0.128	1.308	1.259	1.568	0.309
G2	2.015	0.243	1.976	1.766	2.350	0.584
G3	0.293	0.029	0.286	0.262	0.326	0.064
G4	0.633	0.183	0.629	0.367	0.878	0.511
G5	0.130	0.034	0.114	0.100	0.175	0.075

**Table 5 nanomaterials-12-01951-t005:** Roughness statistical analysis: multiple comparison test.

Group	G2	G4	G5	G1
G4	0.2218	-	-	-
G5	0.0002	0.1901	-	-
G1	1.0000	0.8482	0.0114	-
G3	0.0102	0.5655	0.2827	0.1587

**Table 6 nanomaterials-12-01951-t006:** Microhardness descriptive analysis: mean value, standard deviation, median value, minimum and maximum value, range.

Group	Mean	sd	Median	Mininum	Maximum	Range
G1	61.4	2.51	60	59	65	6
G2	72	1.41	72	70	74	4
G3	64.4	1.14	64	63	66	3
G4	46	1.41	46	44	48	4
G5	47	2.00	48	44	49	5

**Table 7 nanomaterials-12-01951-t007:** Microhardness statistical analysis: multiple comparison test.

Group	G1	G2	G4	G5
G1	-	0.2356	0.2827	0.4412
G3	0.9263	0.6322	0.0524	0.1384
G4	-	0.0007	-	-
G5	-	0.0031	0.6977	-

## Data Availability

The data presented in this study are available on request from the corresponding author.

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
