# Peer review of "Depth of Cure, Hardness, Roughness and Filler Dimension of Bulk-Fill Flowable, Conventional Flowable and High-Strength Universal Injectable Composites: An In Vitro Study"

_nanomaterials, 2022, doi:10.3390/nano12121951_

Round 1
Reviewer 1 Report
1. The polymerization shrinkage of the material is an important factor affecting the repair during the polymerization reaction (line 58), but the results of the polymerization shrinkage of various materials were not seen in this study.
2. In the Scanning Electron Microscope experimental method (line 157), does the morphological observation of the sample only observe the surface of the material? Did the sample cut into two pieces and evaluated from top to bottom?
3. In the Roughness display, the author observes the samples through a 3D laser confocal microscope and calculates the average value. Is there an image on this test?
4. Figure 4. Depth of cure Box-plot. (Line 249), the Y-axis coordinate title is not specified.
5. Figure 7. SEM images, 3000 and 9000 Magnification, G1, 9000X magnification, please capitalize the letters of Magnification.
Author Response
Dear Reviewer,
we would like to thank you very much for the precious and constructive comments you gave us. We really appreciate your precious time.
We addressed every single point you highlighted.
We made the modifications through the text with the "track changes" option on, so they are easy to see.
Once again, thank you very much
Best regards
Dear Reviewer,
we would like to thank you very much for the precious and constructive comments you gave us. We really appreciate your precious time.
Comment 1: The polymerization shrinkage of the material is indeed a fundamental factor: the results of the polymerization shrinkage of various materials are not present in this study because this test was not carried on. ISO 4049 does not specify the obligatoriness to realize this test. In a further study, beside all the test carried on for this paper, also the polymerization shrinkage of the materials could be provided. Thank you
Comment 2: The surface of the material was observed: in mouth the parte of the material which come in contanct with the opposing tooth and which plays a role during mastigation and fonation is the surface part.
Comment 3: two new figures were added as requested: Figure 4 and 5
Comment 4: Figure 4 (new figure 6) was corrected. Thank you
Comment 5: Figure 7 (new figure 9) was corrected. Thank you

Reviewer 2 Report
This is a useful study of the properties of of "bulk-fil" dental composite resins after curing with a light-emitting diode lamp for 20 seconds. It is clearly written and shows that the bulk-fil materials have significantly greater depths of cure than conventional flowable composites or a universal injectable composite. This differs from some other published results, but is explained by reference to differences in the types of cure lamp. Overall, this is an important outcome, and one with clinical relevance.
Despite the high overall quality of the work, a few changes are needed before this paper can be accepted. They are:
Line 20: Capital letter to the word "To";
Line 28, "caliber" should be "caliper". Also, in line 126.
Line 35: Capital letter on the word "The"
Line 42: Capital letter on the word "Bulk".
Line 49: It is not clear how the biocompatibility of these materials has been improved and anyway the word should not be used without mentioning the location within the body. Hence, I would prefer to see this written as "... due to their good biocompatibility in the mouth, esthetics...".
Line 87: The opening sentence is quite clumsy, and should be "The use of bulk-fil composites is still growing, and it is not clear that they are completely effective or have the potential to replace traditional composites."
Line 93: Indent this paragraph.
Line 94: Publications are NOT increasing exponentionally (a term which describes a very specific mathematical function). Hence the word should be replaced with "substantially".
Line 112: The proper name of the organisation is the International Organization for Standardization.
Line 167: Should be "Figure".
Line 178, 193, 211: "outlier" should not have a capital letter.
Line 235/359: There is inconsistency in spelling "polymerization/polymerisation". Either is all right, but it should be one or the other, and the same spelling throughout the paper.
Line 256: Table 1 goes over the page, and should be on one side only.
Line 272: Remove the word "parameter" the first time it appears, i.e. "This is the key parameter...".
Line 290/299/304/306: Do not abbreviate "Table" to "Tab".
Line 309/311/312: Remove initials from references.
Line 317/319: "Benzoyl" should be "benzoyl". Also, it would be useful if the full name of the particular germanium compound could be given here.
Lines 420-491. There is a problem with the references, which are numbered twice, probably because the authors were not expecting the numbering to be put in automatically. Please correct this.
Overall, these are very minor changes, but essential. once they are made the paper will be suitable for publication.
Author Response
Dear Reviewer,
we would like to thank you very much for the precious and constructive comments you gave us. We really appreciate your precious time.
We addressed every single point you highlighted.
We made the modifications through the text with the "track changes" option on, so they are easy to see.
Once again, thank you very much
Best regards
